# Comprehensive Metabolomic Analysis Reveals Dynamic Metabolic Reprogramming in Hep3B Cells with Aflatoxin B1 Exposure

**DOI:** 10.3390/toxins13060384

**Published:** 2021-05-27

**Authors:** Shufeng Wang, Xin Yang, Feng Liu, Xinzheng Wang, Xuemin Zhang, Kun He, Hongxia Wang

**Affiliations:** National Center of Biomedical Analysis, Beijing 100850, China; sfwang@xmail.ncba.ac.cn (S.W.); xyang@xmail.ncba.ac.cn (X.Y.); liuf@proteomics.cn (F.L.); wxz@proteomics.cn (X.W.); zhangxuemin@cashq.ac.cn (X.Z.)

**Keywords:** AFB1, Hep3B, UPLC-ESI-MS/MS, non-targeted metabolomics

## Abstract

Hepatitis B virus (HBV) infection and aflatoxin B1 (AFB1) exposure have been recognized as independent risk factors for the occurrence and development of hepatocellular carcinoma (HCC), but their combined impacts and the potential metabolic mechanisms remain poorly characterized. Here, a comprehensive non-targeted metabolomic study was performed following AFB1 exposed to Hep3B cells at two different doses: 16 μM and 32 μM. The metabolites were identified and quantified by an ultra-performance liquid chromatography-mass spectrometry (UPLC-MS)-based strategy. A total of 2679 metabolites were identified, and 392 differential metabolites were quantified among three groups. Pathway analysis indicated that dynamic metabolic reprogramming was induced by AFB1 and various pathways changed significantly, including purine and pyrimidine metabolism, hexosamine pathway and sialylation, fatty acid synthesis and oxidation, glycerophospholipid metabolism, tricarboxylic acid (TCA) cycle, glycolysis, and amino acid metabolism. To the best of our knowledge, the alteration of purine and pyrimidine metabolism and decrease of hexosamine pathways and sialylation with AFB1 exposure have not been reported. The results indicated that our metabolomic strategy is powerful to investigate the metabolome change of any stimulates due to its high sensitivity, high resolution, rapid separation, and good metabolome coverage. Besides, these findings provide an overview of the metabolic mechanisms of the AFB1 combined with HBV and new insight into the toxicological mechanism of AFB1. Thus, targeting these metabolic pathways may be an approach to prevent carcinogen-induced cancer, and these findings may provide potential drug targets for therapeutic intervention.

## 1. Introduction

AFB1 is a common mycotoxin, with high toxicity and has been classified as a Group I human carcinogen by the International Agency for Research on Cancer (IARC), which has been recognized as a major contaminant in food and animal feed [1]. AFB1 is one of the most potent contributors to the worldwide occurrence of HCC [2] since the liver is the most important target organ of AFB1 [3]. Food contaminated with a high level of AFB1 causes severe health issues, including acute hepatotoxicity to humans and animals, whereas chronic exposure to high levels of mycotoxins leads to increases in the incidence of cancer, including HCC [4,5]. In China, HCC has ranked as the second most frequent fatal cancer since the 1990s [6], and HBV infection is a key factor [7,8]. HBV infection and aflatoxin exposure as independent and interactive risk factors for HCC have been investigated. Their results strongly supported a causal relationship between the presence of the chemical and viral-specific biomarkers and the HCC risk, and also showed that the two risk factors act synergistically [9,10]. HBV is a small hepatotropic DNA virus responsible for some chronic liver diseases in humans and the expression level of liver fatty acid binding protein 1 (FABP1), a key regulator of hepatic lipid metabolism, was elevated in HBV-producing hepatoma cells [11]. HBV infection has been reported to facilitate the metabolism of AFB1 in the liver via the transactivation of the pregnane X receptor (PXR) and induction in CYP3A4 [12]. Moreover, HBV may exacerbate the host response to AFB1 and down-regulated detoxification-related proteins, making HBV-infected hepatocytes more susceptible to AFB1 toxicity. Instead, exposure to AFB1 would also exacerbate and accelerate the disease progression of HBV infection [13]. In a humid and tropical area such as Southern China, many people are chronically infected with HBV while also exposed to AFB1 in their diet [2,14,15]. Compared with patients without these etiological factors, those with HBV infection were much more susceptible to AFB1 and contribute to a high incidence of HCC [16].

Much research has been performed to investigate the toxicological effect and mechanism of AFB1, including metabolomic study. Previous studies revealed that the toxic and carcinogenic effects of AFB1 are attributable to the reacting of its metabolites with cellular macromolecules, such as DNA and proteins [17]. Exposure to AFB1 has been shown to affect many pathogenic events implicated in hepatic lipid homeostasis. For example, exposure to 0.5 and 1 mg kg^−1^ of AFB1 for 7 days results in liver damage and dysregulation of genes associated with lipid metabolism in rats [18]. Oxidative damage was the crucial step in AFB1 induced acute hepatotoxicity in rats, whereas gluconeogenesis and lipid metabolism disorder were found to be the major metabolic effects after acute AFB1 exposure [19]. Zuberi et al. [20] utilized techniques based on nuclear magnetic resonance (NMR) to investigate metabolic changes of zebrafish embryos associated with AFB1 exposure. However, studies addressing the chronic toxicity of AFB1 and comprehensive metabolic effects with AFB1 exposure remain rare since the amount of AFB1 exposed to humans by eating contaminated foods is low and the harmful effect is hard to detect owing to symptom-free at an early stage.

Metabolomics, which is the profiling of metabolites in biofluids, cells, and tissues, can be a powerful tool in illustrating the special metabolic changes within a biological system in response to some challenges, such as physiological or infectious disease, changes in the environment, exposure to toxins, and interactions by drugs or other external stressors [21,22]. Metabolomics has long been recognized as a valuable technique for the identification of a target organ toxicity [23,24,25,26], and the evaluation of chemical toxicities [27,28,29,30]. Metabolomics analyses are routinely performed using several different platforms, including NMR, gas chromatography-mass spectrometry (GC-MS), and liquid chromatography-mass spectrometry (LC-MS). The development of UPLC in recent years has made it possible to achieve higher resolutions, higher sensitivities, and rapid separations [31]. Wang et al. [32] reported the metabolomic changes of dairy cows after AFB1 exposure in multiple biofluids using an NMR-based strategy and revealed that that AFB1 mainly threatens amino acid metabolic pathways. It was reported that AFB1 exposure could change lipid oxidation, carbohydrate, and amino acid metabolism in dairy goats [33]. The metabolome coverage and differential metabolites of NMR-based metabolomic study are limited because of low sensitivity and lack of an ability to identify unknown metabolites [34].

Notably, HBV-infected patients would be more susceptible to the liver injury induced by AFB1. Studies demonstrated 25 μM AFB1 increased liver cancer cell death of Hepa 1-6 and HepG2 by 2.2-fold and 3.0-fold, respectively and the resulting cell debris stimulates HCC tumor growth via an “eicosanoid and cytokine storm” [35,36]. Besides, enhancing the endogenous clearance of debris via eicosanoid regulation, such as dual inhibition of COX-2/sEH, is a strategy to stifle inflammation and thus suppress tumor progression driven by carcinogens. The combined role of HBV infection and AFB1 exposure in metabolic mechanisms is lacking. Thus, it is critically important to investigate the chronic toxic effect of AFB1 exposed to HBV-related HCC cells and its underlying mechanism, which may help to find a preventive strategy. Metabolites change quickly after endogenous or exogenous stimulation; a novel toxic mechanism could be clarified by UPLC-MS-based metabolomics study due to its high sensitivity and high resolution.

In the present study, a non-targeted metabolomics study of Hep3B cells was performed using an ultra-performance liquid chromatography-electrospray ionization tandem mass spectrometry (UPLC-ESI-MS/MS)-based strategy and multivariate statistical analysis to investigate metabolic changes associated with AFB1 exposure. The main objective of this study is to investigate metabolic alterations and explore the mechanism of its chronic toxicity using a high-performance strategy. To improve the identification coverage of metabolites, we performed four analyses per sample by combining reversed phase liquid chromatography (RPLC) and hydrophilic interaction liquid chromatography (HILIC) separations with both positive and negative ionization modes mass spectrometry acquisition. Pathway analysis showed that purine and pyrimidine metabolism, hexosamine pathway and sialylation, fatty acid synthesis and oxidation, glycerophospholipid metabolism, TCA cycle, glycolysis, and amino acid metabolism were significantly changed with AFB1 exposure. The alteration of purine and pyrimidine metabolism and down regulation of the hexosamine pathway and sialylation have not been observed in previous investigations. These findings will provide new insight into the toxic effects and underlying mechanism of AFB1 chronic exposure.

## 2. Results

### 2.1. AFB1 Significantly Inhibits the Proliferation and Promotes the Apoptosis of Hep3B Cells

To select the appropriate concentrations of AFB1 for metabolomic study, we examined cell viability and apoptosis using an MTS kit [37] and flow cytometry in Hep3B cells after exposure to AFB1. Hep3B cells were selected due to the presence of integrated HBV sequences in Hep3B cells and this cell line is commonly used in HBV-related cancer research. As expected, AFB1 significantly inhibits the growth of Hep3B cells (Figure 1A), with apoptosis rates of 8.59% and 14.57% at 16 μM and 32 μM AFB1, respectively (Figure 1B), and with a stronger inhibitory effect (30%, day 6) of 32 μM AFB1 on cell viability than 16 μM (10%, day 6) (Figure 1C). Thus, cells of three groups at day 6 were used for metabolic investigation and designated as DMSO (control), 16 μM, and 32 μM groups.

### 2.2. Metabolic Profiles Based on UPLC-ESI-MS/MS

To characterize the metabolite profiling of Hep3B cells following AFB1 exposure, samples of three biological replicates were analyzed by UPLC-MS in both positive and negative ion modes. To increase the metabolites coverage, two complementary columns (RPLC column and HILIC column) were selected for each sample. Therefore, we analyzed each sample using four modes, i.e., RPLC separation combined with positive ion mode of mass spectrometry (RPLC-POS), RPLC separation combined with negative ion mode of mass spectrometry (RPLC-NEG), HILIC separation combined with positive ion mode of mass spectrometry (HILIC-POS), and HILIC separation combined with positive ion mode of mass spectrometry (HILIC-NEG). For each analysis, three technical replicates (UPLC-MS runs) were performed to increase the reliability of differential metabolites. In total, 108 runs were acquired for all samples except for the quality control (QC) samples, with 27 runs for each analysis mode (RPLC-POS, RPLC-NEG, HILIC-POS, and HILIC-NEG). Acquired data were processed by Progenesis QI and pattern recognition was performed using EZinfo embedded in Progenesis QI. Then metabolic pathway enrichment analysis was performed using MetaboAnalyst (version 5.0). The workflow scheme is depicted in Figure 2A.

The system stability was carried out by injecting a QC every 5 samples during the whole sample batch. The system stability and reproducibility were evaluated by principal component analysis (PCA). The representative PCA score plot (HILIC-NEG) of all experimental samples and QCs is shown in Figure 2B with the first component on the X-axes accounting for about 45.0% and the second component on Y-axes accounting for about 23.8%. A high degree of aggregation was observed in all QCs, suggesting the high stability and reproducibility of the system during the experiment [38].

After peak extraction, alignment, and normalization, a total of 25,903, 11,590, 8174, and 5984 unique ion features (retention time with exact m/z) were obtained by RPLC-POS, RPLC-NEG, HILIC-POS, and HILIC-NEG, respectively. For metabolite identification, two widely used metabolite databases (HMDB and Metlin) were searched. In total, 1227, 665, 825, and 840 metabolites were putatively identified by the four methods respectively, and 2679 metabolites were identified when combining all metabolites from the four methods (Appendix A). The Venn diagram shows the overlap of the metabolites identified by the four methods (Figure 2C). The data indicated that RPLC-POS and HILIC-NEG are more complementary than the other approaches. The identification coverage was 77.2% of total identified metabolites. With the other two approaches, the coverage increased only about 23%. Therefore, we can use RPLC-POS combined with HILIC-NEG methods when instrument time is not enough. The identification rates were from 4.74–15.28% compared with the ion features of corresponding detection methods. The ion features and exact rates are shown in Appendix A. Classes of metabolites detected included glycerophospholipids, fatty acids, organic acids, amino acids, carbohydrates, peptides, acyl glycerides, and sphingolipids. As expected, AFB1 was reliably identified with a high score in three analysis modes, including RPLC-POS, RPLC-NEG, and HILIC-POS, with scores of 48, 55, and 50, respectively (Appendix A).

### 2.3. Multivariate Statistical Analysis

PCA, an unsupervised technique was applied to analyze data obtained by the four analysis methods for revealing the differences between the low-dose group, high-dose group, and control group, respectively. To increase the reliability of the data, PCA analysis was performed for each technical replicate in all four analysis methods. The representative PCA score plot (Figure 3A–D) showed clearer separation of the three groups using RPLC-POS, RPLC-NEG, HILIC-POS, and HILIC-NEG methods, respectively. The result showed that the three groups could be differentiated from the first and second principal components. As shown in Figure 3A, about 42.3 and 29.5% of the variation of the data could be explained by the first and second principal components, respectively. The first two principal components could explain more than 70% of the total variation. The best separation was obtained by HILIC-NEG and the separation of 16 μM group to control group was not clear in RPLC-NEG mode.

To further track specific changes among the three groups, all the data were analyzed using the supervised multivariate partial least squares discriminant analysis (PLS-DA) method and one-way analysis of variance (ANOVA) tests were performed. The representative score plot of the PLS-DA (Appendix A) had similar results as the PCA score plot, showing significant differences between the three groups. To identify which variables were responsible for this separation, the variance importance in the projection (VIP) parameter was used. The VIP plot was obtained through PLS-DA analysis. VIP is a weighted sum of squares of the PLS-DA weight which indicates the contribution to the group discrimination of each metabolite. The higher the VIP value, the higher the corresponding metabolite contribution rate among the three groups. Thus, it is possible to select variables with the most significant contribution for discriminating AFB1 low-dosage, AFB1 high-dosage, and control groups. A one-way ANOVA test was performed to assess the statistical significance. The *p*-values of each ion feature were obtained, and differences were considered to be significant when *p* < 0.05 when compared between every two groups (16/DMSO, 32/DMSO, and 16/32). Fold change (FC) of each metabolite of 16/DMSO, 32/DMSO, and 32/16 were calculated from the mean intensity of three measurements. Finally, the differential ion features were selected based on the following criteria: (1) VIP value ≥ 1; (2) measured in all three technical replicates and *p* < 0.05; (3) average FC ≥ 1.5 or ≤ 0.67 of three technical replicates in each method. Using these requirements, 680, 316, 401, and 319 differential ion features were obtained by RPLC-POS, RPLC-NEG, HILIC-POS, and HILIC-NEG methods, respectively (Appendix A). Among them, the identified metabolites were 161, 70, 107, and 120, respectively. In total, 392 differential metabolites were identified (Appendix A), including glycerophospholipids, acylcarnitines, fatty acids, organic acids, amino acids, amides, saccharides, and various small peptides. The Venn diagram shows the differential metabolite overlap among the four methods (Appendix A). Interestingly, only one metabolite was identified by all four methods, which was adenosine triphosphate (ATP).

### 2.4. Metabolic Pathway Analysis

To explore potential metabolic pathways affected by AFB1, the identified endogenous metabolites (Appendix A) from every comparison (16/DMSO, 32/DMSO, and 32/16) were introduced into the MetaboAnalyst 5.0 for analysis and construction of pathways. The differential metabolites were 225, 322, and 245 in three comparisons, respectively, and the metabolic pathway analysis results are shown in Figure 4A–C. Significantly changed metabolic pathways are labeled. Pathways with an impact value greater than 0.1 or *p* < 0.05 are considered to be significantly changed [39]. Among the 12 pathways affected in the low-dosage group compared to the control group, purine metabolism, alanine, aspartate and glutamate metabolism, pyrimidine metabolism, arginine biosynthesis, D-glutamine and D-glutamate metabolism, and glycerophospholipid metabolism were affected greatly, meeting the two criteria of both *p* < 0.05 and impact ˃ 0.1. The most affected pathway was purine metabolism with an impact of 0.24 and *p* of 2.26 × 10^−5^. The three most strongly affected metabolic pathways were purine metabolism, alanine, aspartate, and glutamate metabolism, and pyrimidine metabolism, which are shown in Appendix A with changed metabolites. As expected, the high-dosage AFB1 group affected 15 metabolic pathways, including 10 pathways changed with low-dosage AFB1 exposure, except for arginine biosynthesis and glutathione metabolism. The additional five pathways were pantothenate and CoA biosynthesis, vitamin B6 metabolism, arginine and proline metabolism, cysteine and methionine metabolism, and arachidonic acid metabolism. We also compared the metabolic pathways that changed between high-dosage and low-dosage AFB1 exposure. In total, thirteen pathways were affected, 12 of them were the same as those pathways affected with AFB1 exposure group compared with control group, with only purine metabolism and pyrimidine metabolism significantly changed with *p* ˂ 0.05. The changed pathways in three comparisons are listed in Table 1. In total, up to 18 pathways changed among the three groups and 5 pathways changed in all three comparisons.

## 3. Discussion

The present study aims to investigate the metabolic consequences of AFB1 exposure in cells and explore the toxicity mechanism of AFB1 by employing a metabolomic strategy. We analyzed metabolic alterations of Hep3B cells extensively by combining RPLC-MS and HILIC-MS analytical procedures in positive and negative ESI modes. Combining the optimized RPLC-MS and HILIC-MS was found to greatly expand the metabolome coverage compared with RPLC alone with 64.02% of new metabolites being identified, enabling the identification of 2679 distinct metabolites in all samples. In total, 392 differential metabolites were obtained by comparing the three groups (16/DMSO, 32/DMSO, and 32/16). Several key metabolic pathways were significantly changed based on pathway analysis by MetaboAnalyst 5.0 and literature findings, including purine and pyrimidine metabolism, glycerophospholipid metabolism, hexosamine pathway and sialylation, fatty acid synthesis and oxidation, TCA cycle, glycolysis, and amino acid metabolism.

### 3.1. Purine and Pyrimidine Metabolism

The pathway analysis of differential metabolites showed that purine metabolism was the most significantly changed pathway in both doses of AFB1 treated cells compared with control cells. As one of the most abundant metabolite classes within a mammalian cell, in addition to the generation of DNA and RNA molecules, purine nucleotides, such as ATP and guanosine triphosphate (GTP) are crucial for providing cellular energy and intracellular signaling, respectively [40]. We found that the levels of fifteen metabolites involved in purine metabolism were changed dramatically, with ten decreased and five increased (Figure 5). The purine metabolism included de novo purine biosynthesis, purine salvage pathway, and purine degradation. These fifteen metabolites were involved in the whole pathway. Xanthosine was the one that decreased most with an FC of 0.02. Inosine, guanosine, and adenosine were among the most decreased metabolites. ATP, adenosine diphosphate (ADP), GTP, and guanosine monophosphate (GMP) were elevated with AFB1 exposure. Overall, the purine metabolism decreased with AFB1 exposure, which may be related to the incorporation of AFB1 into guanine residues of DNA [17]. AFB1 incorporated into guanine residues of DNA was one important toxic effect which caused DNA damage and cell death [41]. However, the systematic purine metabolism alteration caused by AFB1 has not been reported yet. These findings may provide new insight into the toxic mechanism of AFB1 and treatment strategies.

Our results were inconsistent with the previous report, in which inosine and adenosine were markedly elevated in the aqueous liver extracts from rats exposed to AFB1 [42]. This discrepancy may be due to the difference between cell and rat and has merits to be studied further. Investigations suggested that disturbances in purine metabolism caused by genetic polymorphisms could increase the burden of mutagenic deaminated nucleobases in DNA and interfere with gene expression and RNA function [43]. We suspected that the alteration of purine metabolism may be one potential toxic effect of AFB1 leading to HCC.

Interestingly, as the third significantly changed pathway, the pyrimidine metabolism was upregulated in both doses of AFB1, which is opposite to the changing trend of purine metabolism as described above. The levels of 11 metabolites involved in this pathway changed dramatically, with 10 elevated and only uridine diphosphate (UDP) was decreased in a dose-dependent manner (Figure 5). Those upregulated metabolites included uracil, uridine, uridine 5’-monophosphate (UMP), UDP, uridine triphosphate (UTP), cytidine, cytidine triphosphate (CTP), deoxycytidine monophosphate (dCMP), cytidine diphosphate (CDP), and thymidine 5’-triphosphate (TTP). The upregulation of pyrimidine metabolism induced by AFB1 has not been reported yet.

The up-regulation of pyrimidine metabolism agreed with the view that AFB1 exposure may cause the degradation of nucleic acids as these metabolites are the basic structural units for nucleic acids [42]. AFB1-induced up-regulation of genes encoding DNA damage responses including O_6_-methylguanine-DNA methyl transferase, cyclin G1, and thymidylate synthase [44]. The AFB1 metabolite, AFB1-exo-8,9-epoxide, bound with guanine residues of DNA [45] leading to the impairment of DNA and RNA template activity and inhibition of DNA and RNA synthesis [46], which is consistent with the above views. AFB1 incorporated into guanine residues for DNA was one important reason for DNA damage and cell death, which has a similar mechanism as widely used chemotherapeutic agents by inhibiting synthesis and incorporation of nucleotides in DNA to reduce tumor growth, cause DNA damage, and induce cell death [47]. The elevation of the pyrimidine metabolism pathway with AFB1 exposure may be also related to its tumorigenesis. Disruption of the pyrimidine metabolism pathway is associated with tumor cell differentiation and maintaining a continuous proliferative state [47].

It is very interesting that the alteration trend of purine and pyrimidine metabolites in Hep3B cells was inconsistent as basic structural units for nucleic acids following AFB1 exposure in our experiment. We speculated that the binding of AFB1 metabolite with guanine residues of DNA may be one reason for different alteration trends of purine and pyrimidine metabolism. The exact mechanism merits further investigation to test if the toxic effect can be alleviated by interfering with purine and pyrimidine metabolic pathways whencells or animals exposed to AFB1, which could provide some potential drug targets for patient treatment.

### 3.2. Lipid Metabolism and Oxidative Stress

The lipid metabolic pathway collectively displays the largest perturbations with AFB1 exposure. Lipid metabolism includes fatty acid oxidation, fatty acid biosynthesis, and metabolism of phospholipids. One of the most prominent findings in the current investigation was the downregulation of fatty acid oxidation (Figure 5), the β-oxidation which was indicated by the dramatically decreased levels of 20 and 26 acylcarnitines and carnitine in the low and high AFB1 treated groups compared with the control group, respectively (Appendix A). The acylcarnitines were from acetylcarnitine to stearoylcarnitine with decreased FCs from around 0.5 to 0.01 and the decrease was AFB1 dose dependent. As specific substrates of mitochondrial fatty acid β-oxidation, acylcarnitines can facilitate the transfer of long-chain fatty acids from the cytoplasm into mitochondria during the oxidation of fatty acids, and the level of them is the rate-limiting step in fatty acid β-oxidation [48,49]. Acylcarnitines have been reported as differential metabolites in drug-induced hepatotoxicity [50,51], which indicates their essential roles in maintaining normal liver function. It has been reported that the β-oxidation of fatty acid in rat liver tissue with AFB1 exposure was elevated [42], which was inconsistent with our results. The discrepancy may be due to different AFB1 doses or differences between cells and rats, which need to be investigated further.

Acetyl CoA is the start resource for fatty acid biosynthesis, which is originated from citric acid, a metabolite from the TCA cycle. The level of citric acid decreased almost three times in AFB1 exposure cells. The levels of three fatty acid biosynthesis products, palmitic acid, palmitoleic acid, and oleic acid were significantly decreased in low-dose AFB1 treated cells compared to the control cells with FCs of 0.65, 0.63, and 0.44, respectively. These data indicated that fatty acid biosynthesis decreased significantly after cell exposure to AFB1. Our data agreed well with the previous studies in which the expression of fatty acid synthase was decreased measured by PCR in a dose-dependent manner with AFB1 exposure and decrease of citric acid in AFB1 treated rat urine [19,42].

Fatty acids are a very important subclass of fatty acyls. Among the altered metabolites in our study, we found that the levels of many fatty acids were altered significantly with AFB1 exposure, including nine elevated and nine decreased fatty acids, respectively. The increased ones were mainly polyunsaturated fatty acids, including docosahexaenoic acid, arachidonic acid, eicosapentaenoic acid, and tetracosahexaenoic acid. Arachidonic acid is the structural molecular which constitutes phospholipids, whereas phospholipids are the components of the plasma membrane. The significant increase of arachidonic acid in the high-dose AFB1 treated cells was consistent with the phospholipids increasing. This agreed well with a previous study in which high-dose AFB1 treatment caused a hepatic injury, membrane disintegration, and acute cellular swelling [52]. Fatty acids involved in fatty acid biosynthesis were decreased as described above, including palmitic acid, palmitoleic acid, and oleic acid.

Another prominent finding of the present study is the marked elevation in levels of various glycerophospholipids. Phosphatidylcholines (PC), phosphatidylethanolamine (PE), phosphatidylserines (PS), lysophosphatidylcholines (LPC), and lysophosphatidylethanolamine (LPE) belong to the glycerophosphate group. For the twelve differential PC, only four were decreased in the high-dose AFB1 cells. For the fifteen differential PE, only three were decreased in the high-dose AFB1 cells. PS showed similar trends as PC and PE. Four of the thirteen differential PS decreased in the high-dose AFB1 cells. The levels of 15 LPC and 15 LPE were greatly elevated in high-dose AFB1 treated cells compared with the control, with only one LPC decreasing in AFB1 exposed cells. Overall, the levels of glycerophospholipids were elevated with AFB1 exposure. Most of the VIPs of these glycerophospholipids are greater than 2, with the highest being 13.5. Consistently higher levels of other membrane moieties, such as choline and phosphocholine found in cells exposed to AFB1 in the current investigation also agree well with the swelling of cells because choline, phosphocholine, and PC are essential elements for the structural integrity of cell membranes [53].

Oxidative stress followed by lipid oxidation has previously been reported in AFB1 toxicity. Reactive oxygen species (ROS), such as H_2_O_2_ and O_2_, are widely considered to participate in the main mechanism of aflatoxin toxicity [54]. Pathway analysis of our data indicated that glutathione (GSH) metabolism was a significantly changed pathway, with four metabolites being involved, including s-(formylmethyl) glutathione, s-(1,2-dicarboxyethyl) glutathione, cysteine glutathione, and GSH. GSH is a critical cellular antioxidant related to cellular oxidative stress, and GSH modulated drug metabolism is an important mechanism for drug detoxification [55]. Here we observed elevated levels of GSH in cells exposed to low-dose AFB1 with an FC of 1.83 and VIP of 26.2, which means that GSH is an important differential metabolite to AFB1 exposure. GSH is the most abundant antioxidant found in living organisms that acts as a free radical scavenger and a detoxifying agent in cells and it is the most commonly elevated metabolite detected during oxidative stress. Increased GSH production can be induced by common stressors like free radicals, peroxides, lipid peroxides, and heavy metals [56]. The dysregulation of glutathione metabolism and increase of GSH indicated that AFB1 induced oxidative stress and caused mitochondrial dysfunction. An increase of glutathione with AFB1 in rats has been reported before [42]. Besides, disturbances in GSH homeostasis are often found in multiple pathologies, e.g., neurodegenerative disorders, cancer, cystic fibrosis, liver disorders, and diabetes [57]. Therefore, disturbances in GSH homeostasis may be one toxicity mechanism of AFB1 which induces liver disease, including HCC. Furthermore, the other three GSH related metabolites changed significantly, with s-(formylmethyl)glutathione and s-(1,2-dicarboxyethyl)glutathione increased and cysteineglutathione disulfide decreased in AFB1 exposure cells. AFB1 caused mitochondrial dysfunction supported by dramatic downregulation of fatty acid oxidation and dysregulation of the TCA cycle. The significant increase of ROS production because of mitochondrial dysfunction and altered metabolism agreed well with previous findings [58].

Except for the increase of GSH, decreased levels of L-proline and trans-4-hydroxy-proline in cells exposed to high-dose AFB1 were measured with FCs of 0.62 and 0.46, respectively. The decrease of proline and trans-4-hydroxy-proline agreed well with the previous findings [37,59]. Li et al. found that AFB1 and AFM1 induced activation of oxidative reactions in mice and proline concentration was significantly lower in mice treated with AFB1 and AFM1 than the control mice [37]. Li et al. also demonstrated that L-proline alleviates kidney damage by regulating the excessive apoptosis of kidney cells induced by AFB1 and AFM1 in a renal injury model [59]. Lu et al. reported that oxidative damage was the crucial step in AFB1-induced acute hepatotoxicity in rats from general toxicity studies, transcriptomics, and metabolomic profiles [19]. Taken together, our data showed that AFB1 induced oxidative stress and it is one major toxic effect of AFB1, which was consistent with the previous views.

It was reported that lipid oxidation often leads to the generation of hydrogen peroxides, a cause for free radical oxidative damage [54]. Interestingly, our data cannot provide evidence that ROS was produced by lipid oxidation since we observed decreased β-oxidation and increased oxidative stress. Overall, our data indicated that lipid metabolism disorder and oxidative stress were major metabolic effects after AFB1 exposure. However, lipid oxidation was not the cause of oxidative stress due to the dramatic decrease of both fatty acid biosynthesis and β-oxidation. Based on the dramatic decrease of both fatty acid biosynthesis and β-oxidation, activation of the two ways may decrease the toxic effect of AFB1 exposure and benefit HBV patients.

### 3.3. Hexosamine Pathway and Sialylation

Sialylation, the covalent addition of sialic acid to the terminal end of glycoproteins, is a biologically important modification that is involved in embryonic development, neurodevelopment, reprogramming, oncogenesis, and immune responses [60]. In our data, we found that four metabolites in the biosynthesis pathway of sialylation were significantly decreased with AFB1 exposure, including UDP-N-acetyl-glucosamine(UDP-GlcNAc), N-acetyl-D-mannosamine (ManNAc), Neu5Ac, and cytidine 5′-monophosphate N-acetylneuraminic acid (CMP-Neu5Ac) (Figure 5). The nucleotide sugar UDP-GlcNAc is produced by the hexosamine pathway in the cytosol [61]. ManNAc is the metabolic precursor for the synthesis of sialic acid and produces Neu5Ac in the cytosol, which then enters the nucleus to produceCMP-Neu5Ac. Importantly, the VIP of UDP-GlcNAc was 30.6, the largest VIP of all differential metabolites (VIP ≥ 1) and the FC of UDP-GlcNAc was AFB1 dose dependent, with FCs of 0.50 and 0.37 in low-dose and high-dose AFB1 group when compared to control group. These data indicated that the down-regulation of the sialylation biosynthesis pathway may be one key toxic effect of AFB1 exposure. The decrease of the sialylation pathway caused by AFB1 exposure has not been reported yet. Similarly, activating this pathway may alleviate the toxic effect of AFB1 exposure and be helpful for HBV patients.

### 3.4. Amino Acid Metabolism

Pathway analysis showed that several amino acid metabolisms were significantly enriched, including alanine, aspartate and glutamate metabolism, D-glutamine and D-glutamate metabolism, phenylalanine, tyrosine and tryptophan biosynthesis, arginine and proline metabolism, cysteine and methionine metabolism, phenylalanine metabolism, and tryptophan metabolism. In total, 14 metabolites involved in these pathways were significantly changed. Among the 14 differential metabolites, the levels of L-aspartate, *N*-(L-arginino) succinate, S-adenosyl-L-methionine, D-glutamine, L-phenylalanine, 2-aminoacrylic acid, and L-tryptophan were elevated, L-glutamate, *N*-acetylaspartylglutamate, citrate, *N*-acetyl-L-aspartate, D-proline, hydroxyproline, and L-cystathionine were decreased. This implies that AFB1 causes disruptions of amino acid metabolisms. The results were consistent with the previous findings that AFB1 induced a significant increase in the levels of a range of amino acids in rat livers [42]. The decrease of proline level in mice treated with AFB1 and AFM1 has been reported [37], which agrees with our data. Overall, AFB1 exposure induced significant changes in amino acid metabolisms, which play important roles in maintaining cell life.

## 4. Conclusions

In the present study, we comprehensively analyzed the AFB1-induced metabonomic changes in Hep3B cells using UPLC-MS-based metabonomic approaches and metabolic pathway analysis. By combining the orthogonal HILIC-MS and RPLC-MS in both positive and negative ionization modes, we greatly expanded the metabolome coverage. Therefore, hundreds of significant differential metabolites were obtained, and multiple metabolites involved in pathways provided more reliable evidence for perturbed pathways. Our data indicated that AFB1 exposure induced significant metabolic reprogramming (Figure 5), including significantly decreased purine metabolism, hexosamine pathway and sialylation, fatty acid oxidation, fatty acid synthesis, TCA cycle, remarkably increased pyrimidine metabolism, glycerophospholipid metabolism, and perturbed glutathione metabolism, and amino acid metabolism. The findings of alterations of purine and pyrimidine metabolism and the decrease of hexosamine pathways and sialylation provided novel insights into toxicity and related mechanisms of AFB1. These findings may help the research and development of preventive or curative medicines for AFs which are rare in clinical treatment, targeting these metabolic pathways may be an approach to prevent carcinogen-induced cancer.

## 5. Materials and Methods

### 5.1. Instruments and Reagents

Non-targeted metabolomics was conducted using a hybrid Q-TOF mass spectrometer TripleTOF^TM^ 6600 (SCIEX, Foster City, CA, USA) equipped with a TurboV^TM^ electrospray interface and coupled to a Shimadzu Prominence UPLC system LC-30AD (Shimadzu, Kyoto, Japan). The columns ACQUITY^TM^ HSS T3 column (2.1 mm × 100 mm, 1.8 μm) and ACQUITY^TM^ BEH amide column (2.1 mm × 100 mm, 1.7 μm) were purchased from Waters (Waters, Ireland).

An HBV expressing Hep3B cell line used in this study was obtained from the Cell Bank of the Chinese Academy of Sciences (Shanghai, China). Cells were maintained in Dulbecco’s Modified Eagle’s Medium (DMEM) medium supplemented with 10% fetal bovine serum (FBS), 1% penicillin-streptomycin (Sigma) in a 37 °C humidified incubator in the presence of 5% CO_2_. The cells were dislodged with 0.05% trypsin/EDTA solutions (Sigma) when they reached 70–80% confluence. MTS (CellTiter 96^®^ Aqueous One Solution Cell Proliferation Assay, G3580) was purchased from Promega (Madison, WI, USA). AFB1 (purity ≥99%), dimethyl sulfoxide (DMSO), and formic acid (FA) were purchased from Sigma-Aldrich Inc. (Saint Louis, MI, USA). HPLC grade water and acetonitrile were purchased from Fisher Scientific Corp. (Loughborough, UK)

### 5.2. Experimental Method

#### 5.2.1. Cell Treatment and Viability Assay

Five milligrams of AFB1 was dissolved into 2 mL of DMSO and diluted with DMEM to 16 and 32 μM with the final DMSO concentration in the culture medium being ≤0.1% (*v/v*), which had no adverse effect on the cellular parameters tested. 0.1% (*v/v*) DMSO solution was used as control for this assay. For the proliferation assay, cells were plated into 96-well microtiter plates (approximately 1000 cells per well) and incubated overnight at 37 °C. On the following day, Hep3B cells were respectively treated with AFB1 at concentrations of 16 and 32 μM for another six consecutive days. MTS solution was added to each well at a final concentration of 20% and incubated for 1.5 h. The absorbance at 490 nm was measured for growth inhibitory effects on six consecutive days, and the cell growth curves were plotted using GraphPad Prism (version 8) software (GraphPad Software Inc., San Diego, CA, USA). The assays were performed in triplicate and were repeated three times. Error bars indicate SEM, *n* = 9. Hep3B cells were exposed to AFB1 for six consecutive days, which reflects a chronic, long-term, and persistent toxic of the Hep3B cells being exposed to AFB1. To represent a scenario of exposure as realistic as possible, we chose day 6 as the test point for untargeted metabolomics studies.

#### 5.2.2. Flow Cytometry Analysis

Apoptosis was evaluated using the Annexin V-FITC/PI Apoptosis Detection kit (BestBio, Shanghai, China). Briefly, Hep3B cells were grown onto a culture dish incubated with 0, 16, and 32 μM AFB1 for 6 d. The Hep3B cells were trypsinized and washed three times with ice-cold phosphate-buffered saline (PBS), and stained with annexin V-FITC and PI for 15 min and examined under a light microscope equipped with appropriate filters. Apoptotic cells stained with annexin V-FITC showed green fluorescence, and necrotic cells stained with both annexin V-FITC and PI produced red fluorescence as well as green fluorescence. Percentages of cells in the different phases of the cycle were measured by flow cytometric analysis of propidium iodide-stained nuclei using a Flow Cytometer (BD Accuri^TM^ C6 Plus) (BD, Franklin Lakes, NJ, USA). Proportions of apoptotic cells were indicated as a percentage of the sub-G1 and sub-G2 fraction in FACS analysis.

#### 5.2.3. Metabolite Extraction from Hep3B Cells

For non-targeted metabolomic analysis, Hep3B cells were pooled into culture dishes (15 cm in diameter) at a density of 1 × 10^6^ cells per dish. After being cultured for 24 h, the culture mediums were replaced with 0.1% (*v/v*) DMSO (control group) or medium containing AFB1 at 16 and 32 μM (treatment group). Then cells were exposed to AFB1 for six consecutive days and collected for metabolites extraction. All cells were washed three times with PBS buffer, then quenched by ice-cold methanol/water (80:20, *v/v*) and incubated the plates at −80 °C overnight, the cells were scraped with a cell scraper, transferred to pre-cooling centrifuge tubes, intermittently vortex for 3 min, and centrifuged (14,000 rpm, 4 °C) for 20 min. The supernatants were speedVac/lyophilized to pellets and frozen at −80 °C before UPLC-ESI-MS/MS analysis [62]. Cell culture and metabolite extraction were repeated in triplicate.

#### 5.2.4. UPLC-ESI-MS/MS Analysis

The metabolite profiling of all samples was acquired by an ultrahigh-performance liquid chromatography system coupled to a hybrid Q-TOF mass spectrometer (UPLC-QTOF-MS) in both positive and negative ion modes. To increase the metabolome coverage, the metabolite separation was performed on an HSS T3 column and a BEH amide column, respectively. For the RPLC separation (T3 column), the mobile phases were (A) water with 0.1% FA and (B) acetonitrile with 0.1% FA. The gradient was set as follows: 0–1.5 min, 1% B; 1.5–13.0 min, 1–99% B; 13.0–16.5 min, 99% B; 16.6–20.0 min, 1% B. For the HILIC separation (BEH amide column), the mobile phases were (C) 10% water/acetonitrile with 10 mM NH_4_AC and 0.1% FA (D) water with 10 mM NH_4_AC and 0.1% FA. The gradient was set as follows: 0–1.0 min, 5% D; 1.0–12.0 min, 5–55% D; 12.1–15.0 min, 55% D; 15.1–20.0 min, 5% D. The flow rate was 0.3 mL min^−1^ and sample injection volume was 3 µL for both columns. The column temperature was maintained at 40 °C and the sample chamber temperature was at 6 °C in both positive ion mode and negative ion modes. To verify the chromatographic stability during the measurement of the samples, quality control (QC) was inserted into every 5 samples in the analysis batch to ensure the accurate and orderly analysis of samples [63]. QC was a mixed sample by taking an equal volume (based on the number of samples) from each sample.

TripleTOF^TM^ 6600 mass spectrometry was operated in data-dependent analysis (DDA) mode in both positive and negative modes. Twelve MS2 spectra were acquired with each full MS1 scan. The scan range of precursor ion and fragment ion m/z were both set as 100–1250 Da. The accumulation time for MS1 and MS2 was 150 ms and 30 ms, respectively. The collision energy (CE) voltage was in series set at 15 V, 30 V, and 45 V testing parameters: gas1 and gas2 were 50 Pa; curtain gas was 35 Pa. The desolvation gas temperature and source temperature were both 500 °C. Ion spray voltage floating (ISVF) was 5 kV (ESI+) or −4.5 kV (ESI−); declustering potential (DP) were 80 V (ESI+) or −80 V (ESI−); CE was 30 ± 10 V. Three technical replicates (three injections) were performed for each sample in the four analysis modes.

#### 5.2.5. Data Processing and Analysis

In this study, data processing was performed using the Progenesis QI (version 3.0.3) software (Waters, Milford, MA, USA) and EZinfo (version 3.0) (Waters, Milford, MA, USA) as described by Cheng et al. [64]. Firstly, the original data was preprocessed by Progenesis QI for peak detection and comparison to acquire a list of data information about retention time, molecular weight, and normalized peak area of each sample. Afterward, the resulting data were imported to EZinfo for multivariate pattern recognition analysis, including unsupervised PCA and supervised PLS-DA after mean-centering and Pareto scaling. Variable importance in projection (VIP) value, generated in PLS-DA processing, represents the contribution to the group discrimination of each metabolite [38]. In this study, VIP ≥ 1 was regarded as one criterion of differential features or metabolites. The statistical significance of mean values was tested by one-way analysis of variance (ANOVA) using EZinfo, and differences were considered to be significant when *p* < 0.05.

Metabolite identification was achieved by searching the database HMDB which was downloaded from the HMDB website (https://www.hmdb.ca/, accessed on 12 April 2021) and Metlin which was purchased from Waters Company (Waters, Milford, MA, USA). The identification score from QI is based on accurate mass, fragment pattern, and isotope similarity (20 for each item and the highest score is 60). The metabolites with QI scores of ≥40 were accepted as putatively identified and also manually checked by comparing the measured tandem spectra with the experimental (Metlin) or predictable (HMDB) tandem spectra to confirm their reliable identification. The mass tolerance was 12 ppm for both precursor and fragment ions. The database search was performed before pattern recognition analysis.

#### 5.2.6. Metabolic Pathway Analysis

To identify the most relevant pathways involved in AFB1 exposure, differential metabolites from each two comparisons (low-dose AFB1 to control defined as 16/DMSO, high-dose AFB1 to control defined as 32/DMSO, and high-dose AFB1 to low-dose AFB1 defined as 32/16) were submitted for pathway analysis in the web tool MetaboAnalyst 5.0, which combines results from powerful pathway enrichment analysis with the pathway topology analysis [65].

## Figures and Tables

**Figure 1 toxins-13-00384-f001:**
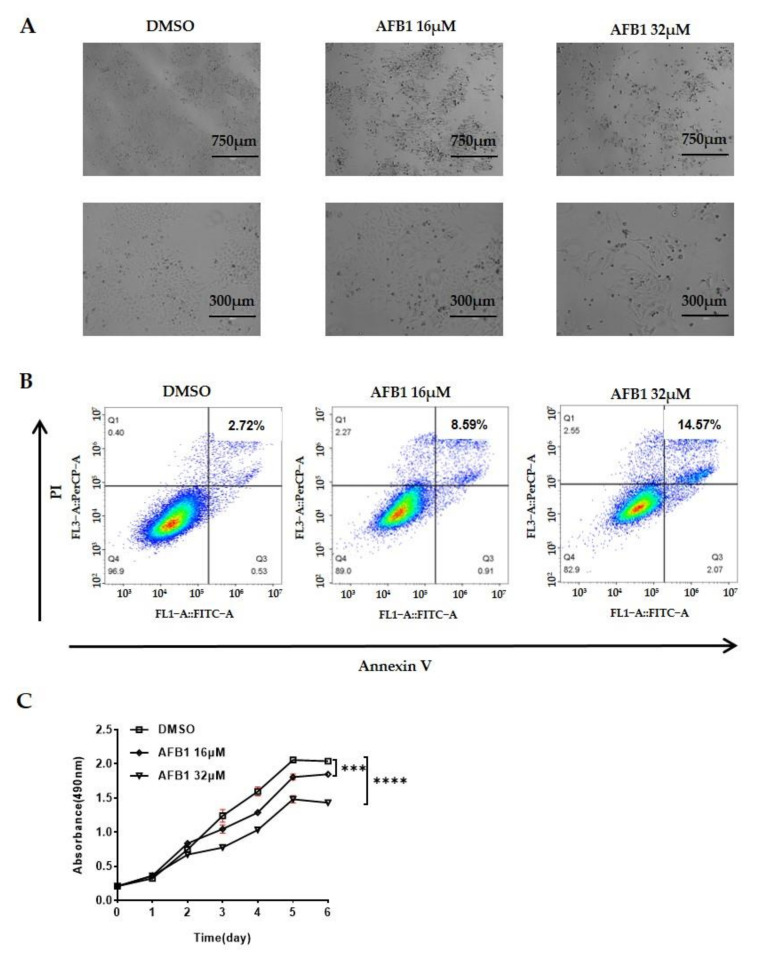
Growth inhibition of Hep3B cells following 16 μM and 32 μM AFB1 treatment for 6 days. (**A**) Representative images illustrate the growth inhibition of Hep3B cells following AFB1 treatment. (**B**) The apoptosis rate of Hep3B cells following AFB1 treatment. (**C**) Cell viability of Hep3B cells following AFB1 treatment. 0.1% (*v/v*) DMSO treated cells as control group. Data are represented as mean ± SEM. ***, *p* < 0.001, ****, *p* < 0.0001, as assayed by unpaired Student’s *t* test.

**Figure 2 toxins-13-00384-f002:**
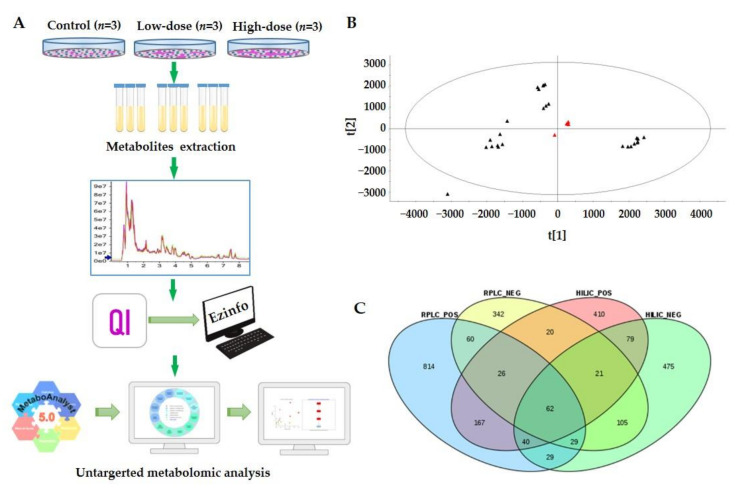
Non-targeted metabolomic profiling of AFB1 treated Hep3B cells by an UPLC-ESI-MS/MS-based strategy. (**A**) The schematic metabolomic workflow of Hep3B cells treated with two doses of aflatoxin B1 using an UPLC-ESI-MS/MS-based approach. (**B**) PCA score plot of QC samples and all experimental samples in HILIC-NEG mode. (**C**) Venn diagram depicting overlap of identified metabolites detected in all samples using four combined modes (RPLC-POS, RPLC-NEG, HILIC-POS, and HILIC-NEG).

**Figure 3 toxins-13-00384-f003:**
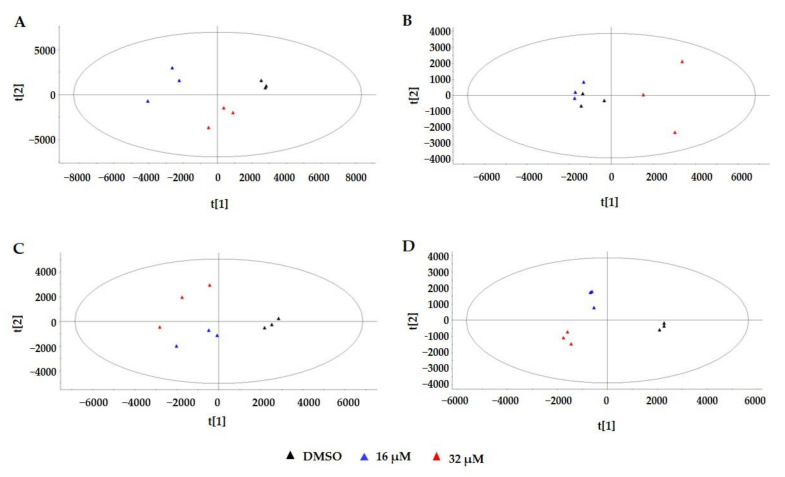
PCA score plots of all metabolic features obtained from three biological replicates of AFB1 treated groups and control groups by RPLC-POS (**A**), RPLC-NEG (**B**), HILIC-POS (**C**), and HILIC-NEG (**D**) modes, respectively. Samples with different treatments in (A–D) are illustrated at the bottom of the figure. DMSO: dimethyl sulfoxide; 16 μM: 16 μM AFB1; 32 μM: 32 μM AFB1.

**Figure 4 toxins-13-00384-f004:**
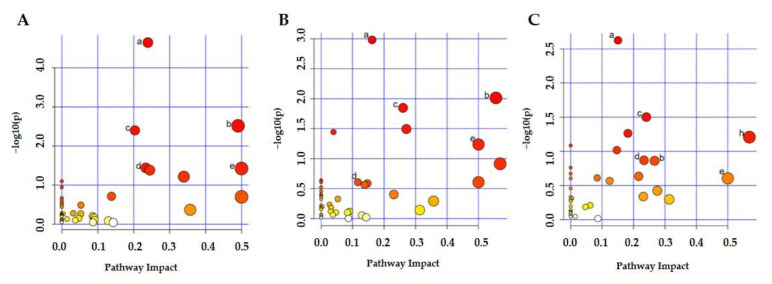
Metabolic pathway results obtained from differential metabolites of three group comparisons, (**A**) 16 μM vs. DMSO, (**B**) 32 μM vs. DMSO, and (**C**) 32 μM vs. 16 μM. The metabolome view shows matched pathways arranged by *p*-values from pathway enrichment analysis (Y-axis) and pathway impact values from pathway topology analysis (X-axis). Node color and radius are based on the *p*-value and pathway impact value, respectively. Labeled nodes denote significantly changed pathways with *p* < 0.05 or pathway impact ˃ 0.1. The five common pathways were (a) purine metabolism, (b) alanine, aspartate and glutamate metabolism, (c) pyrimidine metabolism, (d) D-glutamine and D-glutamate metabolism, and (e) glycerophospholipid metabolism. The detailed information for each significantly changed pathway is listed in Table 1.

**Figure 5 toxins-13-00384-f005:**
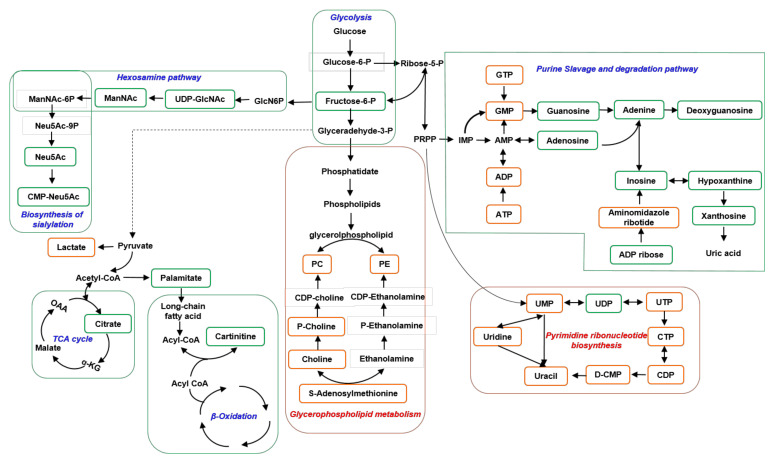
Schematic depiction of the significantly changed key metabolic pathways following AFB1 exposure and their connections. All altered pathways included the purine and pyrimidine metabolic pathways, fatty acid biosynthesis and β-oxidation, TCA cycle, glycolysis, hexosamine pathway, and biosynthesis of sialylation. Significantly downregulated metabolites are shown in green rectangular box, significantly upregulated metabolites are shown in orange rectangular box. Abbreviations (other than standard nucleotide or amino acid abbreviations): P: phosphate; PRPP: phosphoribosyl pyrophosphate; PC: phosphatidylcholine; PE: phosphatidyl ethanolamine; CoA: coenzyme A: aKG, a-ketoglutarate; OAA: oxaloacetic acid; GlcN6P: glucosamine 6-phosphate; UDP-GlcNAc: UDP-N-acetyl-glucosamine; ManNAc-6P: N-acyl-D-mannosamine 6-phosphate; Neu5Ac-9P: N-acylneuraminate 9-phosphate.

**Table 1 toxins-13-00384-t001:** Significantly changed metabolic pathways affected by AFB1 exposure.

Pathway Number	Total	Hits			*p*-Value			Impact		
		16/DMSO	32/DMSO	32/16	16/DMSO	32/DMSO	32/16	16/DMSO	32/DMSO	32/16
1	65	13	12	9	2.26 × 10^−5^	1.05 × 10^−3^	2.38 × 10^−3^	0.24	0.16	0.15
2	28	6	6	3	3.05 × 10^−3^	9.69 × 10^−3^	/	0.49	0.56	0.27
3	39	7	7	5	3.99 × 10^−3^	1.42 × 10^−2^	3.15 × 10^−2^	0.20	0.26	0.24
4	14	3	/	2	3.62 × 10^−2^	/	/	0.23	/	0.23
5	6	2	2	1	3.78 × 10^−2^	/	/	0.50	0.50	0.50
6	36	5	6	3	4.15 × 10^−2^	3.21 × 10^−2^	/	0.24	0.27	0.22
7	19	/	4	3	/	3.60 × 10^−2^	/	/	0.04	0.18
8	9	/	2	2	/	/	/	/	0.57	0.57
9	38	/	4	4	/	/	/	/	0.26	0.10
10	39	/	/	3	/	/	/	/	/	0.12
11	28	4	/	2	/	/	/	0.34	/	0.28
12	33	/	3	2	/	/	/	/	0.23	0.23
13	36	/	2	2	/	/	/	/	0.31	0.31
14	15	2	2	/	/	/	/	0.14	0.14	/
15	4	1	1	/	/	/	/	0.20	0.25	/
16	10	1	1	/	/	/	/	0.36	0.36	/
17	30	1	1	/	/	/	/	0.13	0.13	/
18	41	1	1	/	/	/	/	0.14	0.14	/

*p* < 0.05 or impact ˃ 0.1 indicate significantly enriched pathways. The 18 total pathways were 1. purine metabolism; 2. alanine, aspartate and glutamate metabolism; 3. pyrimidine metabolism; 4. arginine biosynthesis; 5. D-glutamine and D-glutamate metabolism; 6. glycerophospholipid metabolism; 7. pantothenate and CoA biosynthesis; 8. vitamin B6 metabolism; 9. arginine and proline metabolism; 10. fatty acid degradation; 11. glutathione metabolism; 12. cysteine and methionine metabolism; 13. arachidonic acid metabolism; 14. nicotinate and nicotinamide metabolism; 15. phenylalanine, tyrosine, and tryptophan biosynthesis; 16. phenylalanine metabolism; 17. inositol phosphate metabolism; 18. tryptophan metabolism. Total is the total number of compounds in the pathway; Hits is the actual matched number from the user uploaded data; Raw *p* is the original *p*-value calculated from the enrichment analysis; Impact is the pathway impact value calculated from pathway topology analysis. 16/DMSO, 32/DMSO, and 32/16 refer to the low-dose AFB1 group compared to control group, the high-dose AFB1 group compared to control group, and the high-dose AFB1 group compared to low-dose AFB1 group, respectively.

## Data Availability

The data presented in this study are available in the article here.

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
