# Peer review of "Comprehensive Metabolomic Analysis Reveals Dynamic Metabolic Reprogramming in Hep3B Cells with Aflatoxin B1 Exposure"

_toxins, 2021, doi:10.3390/toxins13060384_

Round 1

Reviewer 1 Report

In this paper, the authors analyze the effect of aflatoxin B1 exposure on metabolic reprogramming in Hep3B cells. Altered metabolism in cancer has been recently reconised as an important hallmark of cancer. The authors indicate principal pathways significantly modified, including purine and pyrimidine metabolism, hexosamine pathway and sialylation. However there are some concerns to be discussed:

  1. The study is based only on human liver cancer cells, there are on primary hepatocytes. Please comment.
  2. Representative images of Hep3B cells treated with AFB1 are needed to illustrate better growth inhibition of Hep3B cells following 16μM and 32μM AFB1 treatment for 6 days.
  3. What about the apoptosis rate in this experimental study?
  4. What are the levels before and after treatment of CYP3A4 and CYP1A2, that play central roles in the bioactivation of AFB1 in human?

Author Response

Dear Reviewer:

Thank you for the timely comments of our manuscript, we respond to your questions point-by-point. We look forward to hearing your thoughts on our resubmission.

Reviewer 2 Report

Should be reviewed for minor grammatical errors

Author Response

(The authors gave the same response as above.)

Reviewer 3 Report

This article describes a methodology study in cells for hepatitis B infection showing interesting results on affected metabolic pathways by UPLC-MS analysis.

The introduction and methodology (chromatographic-mass spectrometric and statistical/data analysis) is well described. Moreover, the results are well exposed and discussed for toxicological mechanism of AFB1. I find this paper interesting to be published, however, some minor changes and issues would improve the paper:

- Introduction, page 3, line 104. Explain very shortly to the readers why Hep3B cells are selected? Are not there any other type to be studied for HBV-AFB1?

- Results, page 3, line 127. Do you mean DMSO group as blank against AFB1 treatment?

- Results, figure 3 and supplemental material. According to the results on hit / features and PCA / PLS, it seems that RPLC-POS and HILIC-NEG are more complete than the others approaches… do you dare to discuss a bit about these? If you think that the four approaches are totally complementary, please, explain in the text or in Conclusion (page 12).

- Discussion 3.1, page 9, line 337. Explain a bit which is the purpose of this further investigation. How could the exact mechanism help the patients?

- Conclusion, page 12. Last but not least, the last sentence of the abstract and the last sentence of the conclusion are the same. How could this research help in developing medicines? Please, explain briefly (as a challenge) in each section of the discussion in which way the pathway studied could help clinical treatment for real HBV patients.

For example, it is known that cells AFB1 exposure affects to oxidative stress and lipid oxidation. Beside the affected metabolic pathway in Hep3B cells, what extra information does your work give to the clinical practice?

Author Response

(The authors gave the same response as above.)

Round 2

Reviewer 1 Report

The authors answered thoroughly the reviewer's comments.